# miR614 Expression Enhances Breast Cancer Cell Motility

**DOI:** 10.3390/ijms22010112

**Published:** 2020-12-24

**Authors:** Tuyen T. Dang, Alec T. McIntosh, Julio C. Morales, Gray W. Pearson

**Affiliations:** 1Department of Neurosurgery and Stephenson Cancer Center, University of Oklahoma Health Science Center, 1122 NE 13th St., Oklahoma City, OK 73117, USA; Tuyen-Dang@ouhsc.edu (T.T.D.); Julio-Morales@ouhsc.edu (J.C.M.); 2Simmons Comprehensive Cancer, University of Texas, Southwestern Medical Center, 6001 Forest Park Rd., Dallas, TX 75390, USA; 3Lombardi Comprehensive Cancer Center, Department of Oncology, Georgetown University, 3970 Reservoir Rd. NW, Washington, DC 20057, USA; atm74@georgetown.edu

**Keywords:** miR614, TAPT1, Miro1

## Abstract

Using a data driven analysis of a high-content screen, we have uncovered new regulators of epithelial-to-mesenchymal transition (EMT) induced cell migration. Our results suggest that increased expression of miR614 can alter cell intrinsic gene expression to enhance single cell and collective migration in multiple contexts. Interestingly, miR614 specifically increased the expression of the EMT transcription factor Slug while not altering existing epithelial character or inducing other canonical EMT regulatory factors. Analysis of two different cell lines identified a set of genes whose expression is altered by the miR614 through direct and indirect mechanisms. Prioritization driven by functional testing of 25 of the miR614 suppressed genes uncovered the mitochondrial small GTPase Miro1 and the transmembrane protein TAPT1 as miR614 suppressed genes that inhibit migration. Notably, the suppression of either Miro1 or TAPT1 was sufficient to increase Slug expression and the rate of cell migration. Importantly, reduced TAPT1 expression correlated with an increased risk of relapse in breast cancer patients. Together, our results reveal how increased miR614 expression and the suppression of TAPT1 and Miro1 modulate the EMT state and migratory properties of breast cancer cells.

## 1. Introduction

The activation of epithelial-to-mesenchymal transition (EMT) programs confers tumor cells with the ability to invade [1]. These invasive cells have the potential to intravasate into blood and lymphatic vessels and disperse to new organs [2]. This dissemination can lead to metastatic growth and a significant reduction in the odds of patient survival [3]. EMTs are normally activated to promote migration during organismal development or in response to tissue damage [4]. In tumors, EMT programs can be initiated by genetic abnormalities and extrinsic signals, such as TGFβ, which are produced by immune and mesenchymal cells recruited to the tumor microenvironment [5]. EMTs are coordinated by transcription factors (EMT-TFs) that induce mesenchymal genes, such as the microfilament protein vimentin, extracellular matrix (ECM) targeting proteases and regulators of the cytoskeleton that enhance the rate of cell migration [6,7]. EMT-TFs also suppress epithelial traits, such as the cell–cell adhesion protein E-cadherin [8,9,10,11]. Rather than being a single conserved program, EMTs are highly variable and have distinct invasive phenotypes [1]. For example, depending on the nature of the EMT state, cells can engage in single cell migration or collective migration [12,13,14]. EMT attributes also influence migration rate and the relative ability of cells to reorganize the ECM [15]. Thus, there is variability in EMT programs, and this heterogeneity influences the invasive properties of tumor cells. However, while it is becoming clear that there are many types of EMTs, the factors that specify EMT properties are only beginning to become unraveled. Therefore, identifying regulatory programs that influence EMT induced invasion can improve prognostic measures and assist in specifying treatment options [16].

MicroRNAs (miRNAs) influence the EMT state of tumor cells and specify the mode of tumor invasion. miRNAs are not strictly suppressors or activators of EMTs [17]. For example, the miR200 family limits the extent of EMTs by suppressing the EMT-TFs ZEB1 and ZEB2, whereas miR221/222 enhances ZEB2 expression by suppressing TRPS1 [18]. To better understand how miRNAs influence EMT induced migration, we previously integrated miRNA expression analysis with a high-content screen in which we tested how 879 miRNA mimics influenced wound closure in a breast cancer cell line model [13]. From this, we discovered that miR203 was highly expressed in estrogen receptor positive breast cancer cells and restricts the induction of an EMT program by targeting the transcription factor ΔNp63 [13]. Conversely, we determined that miR205 is highly expressed in a subset of basal-like breast cancer cell lines and enhances the rate of migration [13]. Interestingly, miR205 restricts the induction of EMT program components that promote the reorganization of ECM fibers into parallel tracks that facilitate the initiation of collective invasion [15]. Thus, miR205 confers a hybrid EMT state in which invasion is dependent on the reorganization of the ECM by sibling tumor cells that either lack miR205 expression or by recruited fibroblasts. These results, and those of other groups, demonstrate that miRNAs shape the nature of EMTs [1,17]. Importantly, this EMT modulating function of miRNAs contributes to cell lineage specific responses to EMT inducing signals and creates the potential for the integration of EMT programs with factors regulating miRNA expression.

To better define factors that influence EMT induced migration, we focused our investigation on miRNAs that provoked the greatest enhancement of wound closure in our previously performed high-content screen [13]. We discovered that miR614 enhanced wound closure, the spontaneous motility of subconfluent cells and transwell migration. miR614 had previously been shown to be more highly expressed in African American triple-negative breast cancer (TNBC) patients [13]. However, an understanding of miR614 functions in breast cancer or any biological processes was limited. Thus, we focused on defining how miR614 regulates gene expression. We determined that miR614 specifically induces the EMT-TF Slug, yet does not suppress epithelial traits, including the expression of E-cadherin. A comprehensive analysis of gene expression changes revealed an miR614 regulated gene expression signature. Importantly, functional testing of 25 genes suppressed by miR614 uncovered the outer mitochondrial membrane small GTPase Miro1 and the putative transmembrane protein TAPT1 as miR614 targets that restrict migration. Indeed, suppression of Miro1 or TAPT1 was sufficient to increase Slug expression and promote migration. Moreover, reduced TAPT1 expression correlated with worse odds of relapse-free survival in breast cancer and non-small cell lung cancer (NSCLC) patients. Together, our results reveal that miR614 and the suppression of Miro1 and TAPT1 potentiate Slug expression and EMT induced migration.

## 2. Results

### 2.1. miR614 Enhances Cell Motility

To advance our understanding of factors that influence cell migration, we evaluated the results of a screen in which we determined how miRNA mimic transfection impacted the closure of experimentally introduced wounds [13]. Our previous analysis had used endogenous miRNA expression to prioritize investigating miR203 as a suppressor and miR205 as an enhancer of migration [13]. Here, we focused on miRNA mimics that induced the greatest enhancement of wound closure. Mimics for miR1276 and miR614 were the top inducers of wound closure among the 879 mimics tested in our previously reported screen [13] (Appendix A). Further testing showed that the miR614 and miR1276 mimics consistently enhanced wound closure (Figure 1A). The wound closure assay duration was 24 h, reducing the potential contribution of cell proliferation to the wound closure phenotype. Indeed, miR614 and miR1276 did not enhance the relative cell number compared to control conditions, as determined by overall fluorescent signal in the transfected cells (Figure 1B). In addition, time-lapse imaging of cells during wound closure showed that miR614 and miR1276 increased the speed of cell movement and displacement of cells (Figure 1C and Appendix A). The wound closure of MCFDCIS cells involves the collective movement of cohesive cells (Figure 1A,C). To determine whether miR614 and miR1276 enhance other types of cell motility, we performed time-lapse imaging of subconfluent cells. This allowed us to track the spontaneous movement of individual cells, which included single-cell migration. The miR614 mimic enhanced the speed and displacement of subconfluent cells, whereas miR1276 only modestly increased the speed of movement and did not influence the overall displacement of cells (Figure 1D and Appendix A). These results indicate that miR614 can generally enhance cell motility while miR1276 specifically influences wound closure (Figure 1E). Thus, we prioritized miR614 for further investigation.

### 2.2. miR614 Increases Slug Expression but Does Not Suppress Epithelial Traits

miR614 had not been implicated in the regulation of cell motility and little was understood regarding miR614 function in any context. Therefore, we focused on determining how the miR614 mimic altered gene expression. Consistent with previous analysis of miRNA mimic induced gene expression changes by us and others [13], the miR614 mimic induced the expression of 829 genes and suppressed the expression of 704 genes by at least 2-fold 48 h after transfection into the MCFDCIS cells (Figure 2A and Appendix A). The transcription factor ΔNp63 is essential for MCFDCIS migration [13]. However, miR614 did not increase ΔNp63 expression (Figure 2B). Next, we evaluated the expression of the ΔNp63 induced genes Slug, Axl and FAT2, which we previously determined were required for MCFDCIS migration [13,19]. Interestingly, miR614 did enhance the expression of Slug (Figure 2B), indicating that miR614 can influence the EMT state of MCFDCIS cells. Slug can increase the expression of additional EMT-TFs [1]. However, the expression of canonical EMT-TFs Snail, TWIST1, ZEB1 and ZEB2 were unchanged by miR614 expression (Figure 2C). Epithelial traits, such as the expression of the cell–cell adhesion protein E-cadherin, can be suppressed as part of EMT programs to increase the rate of cell motility [1]. However, E-cadherin was not suppressed by miR614 expression (Figure 2C). miR614 also did not suppress the expression of the epithelial marker EpCAM (Figure 2C). Consistent with these results, immunostaining showed that miR614 did not alter E-cadherin expression or subcellular localization in cell monolayers or in migrating cells proximal to the wound edge, despite the reduced cell–cell cohesion indicated by gaps between cells relative to what is observed in the confluent monolayers (Figure 2D,E). These results suggest that miR614 does not increase cell motility through altering the nature of cell–cell contacts relative to what was observed in control wounds, consistent with the ability of miR614 to increase the migration rate of subconfluent MCFDCIS cells. To further test this possibility, we determined the ability of miR614 to increase the migration of individual MDAMB231 cells through the pores of transwell filters. MDAMB231 cells do not express detectable levels of E-cadherin [13]. Indeed, transfection with miR614 increased the number of cells that migrated through the transwell filter, indicating that miR614 promotes migration, at least in part, through processes beyond the regulation of cell–cell adhesion (Figure 2F). Together, our results indicate that miR614 specifically induces Slug expression without altering the expression of other canonical regulatory features of EMT programs (Figure 2G).

### 2.3. Evaluation of miR614 Target Genes as Regulators of Cell Motility

miRNAs suppress gene expression through binding to seed sequences in 3′ untranslated regions (UTRs). Thus, the induction of Slug expression by the miR614 mimic was indirect, with miR614 suppressing genes that restrict Slug expression (Figure 3A). In addition, miR614 was potentially suppressing genes that restrict migration through processes other than the control of Slug (Figure 3A). Therefore, we sought to identify genes that restrict migration and are suppressed by the miR614 mimic. To prioritize genes for investigation, we identified genes that were suppressed in both the MCFDCIS and MDAMB231 cells (Figure 2A and Figure 3B,C and Appendix A). Consistent with our analysis of MCDCIS cells, miR614 suppressed 1502 genes when transfected into the MDAMB231 cells (Figure 3B and Appendix A). There was an overlap of 101 genes that were suppressed in both the MCFDCIS and MDAMB231 cells (Figure 3C and Appendix A). From this set, we further analyzed the function of 25 genes that were above a minimum expression level in control cells and suppressed by miR614 (Figure 3D). The extent of MCFDCIS wound closure was determined after transfection with siRNAs targeting the miR614 targets in a one-gene/one-well format (Figure 3E). Using a relative wound area of <0.5 as a threshold (smaller wound area indicates more rapid migration), we prioritized the siRNA pools targeting the small GTPase Miro1 and the transmembrane protein TAPT1 for further investigation. We do note that other genes had a relatively more modest but consistent enhancement of wound closure, indicating that they potentially also functioned as migration suppressors targeted by miR614 as well. The remaining siRNAs had limited impact or suppressed wound closure. Suppression of wound closure was potentially a consequence of off-target silencing of gene expression, toxicity or distinct phenotypes resulting from more complete silencing of target gene expression by siRNAs rather than partial suppression of gene expression by miR614.

### 2.4. The miR614 Targets Miro1 and TAPT1 Suppress Migration and Slug Expression

Specific testing of siRNAs targeting Miro1 and TAPT1 in additional experiments showed a reproducible enhancement of wound closure (Figure 4A). Miro1 and TAPT1 expression were depleted by their respective siRNA pools (Figure 4B). To reduce the risk that the phenotype was a consequence of off-target gene silencing, we analyzed wound closure after transfection with new distinct siRNA pools targeting Miro1 and TAPT1. Both the new siRNA pools enhanced wound closure and suppressed target expression, strongly indicating that Miro1 and TAPT1 suppression is the cause of the increased rate of wound closure (Figure 4C,D). To determine whether the suppression of Miro1 and TAPT1 potentially contributed to the regulation of Slug expression, we analyzed Slug mRNA levels in MCFDCIS cells transfected with Miro1 or TAPT1 siRNA. Suppression of either Miro1 or TAPT1 was sufficient to increase Slug expression (Figure 4E). These results identify two targets of miR614 suppression that regulate Slug expression. Using the Targetscan algorithm, a predicted seed sequence was identified in the 3′ UTR of Miro1 for miR614 targeting, but not TAPT1 (Figure 4F), indicating that miR614 has the potential to directly suppress Miro1 expression. However, TAPT1 regulation is likely indirectly regulated by an undefined direct target of miR614. These results indicate that the transfection of cells with the miR614 mimic influences gene expression through direct and indirect mechanisms. This direct and indirect control of gene expression influences the expression of Miro1 and TAPT1, which restricts EMT-TF Slug levels and limits the rate of wound closure (Figure 4G).

### 2.5. High TAPT1 Expression Correlates with Improved Odds of Disease-Free Survival

Identifying traits that stratify patients into outcome groups has the potential to improve the application of current and future systemic therapy. Our results suggested that the miR614 targets Miro1 and TAPT1 regulate cell migration, a key property of cells that metastasize. Thus, to determine whether Miro1 and TAPT1 could be prognostic factors, we determined the correlation between Miro1 and TAPT1 expression and disease-free survival. Indeed, higher TAPT1 expression correlated with an improved chance of disease-free survival in a pan-analysis of breast cancer patients (Figure 5A). These results are consistent with our functional results showing that reduced TAPT1 expression promotes cell migration, which could promote metastasis. Notably, a correlation between high TAPT1 expression and improved odds of survival was also detected when analyzing non-small cell lung cancer (NSCLC) patients, suggesting that TAPT1 may generally influence tumor progression. In contrast, Miro1 expression did not correlate with disease-free survival in breast cancer patients (Appendix A). This lack of correlation with patient outcome could be due to a number of factors, including Miro1 being suppressed as part of other miR614 independent programs that fail to increase migration, or having collateral costs such as restricting colonizing growth. Thus, our analysis has revealed a new potential biomarker, TAPT1, that classifies patients into distinct outcome groups.

Our combined findings suggest that reduced TAPT1 expression, either in response to increased miR614 expression or other regulatory mechanisms, increases the expression of Slug and promotes cell migration (Figure 5B). Similarly, the suppression of Miro1 also induced Slug expression and migration (Figure 5B). Importantly, increased cell migration in response to elevated miR614 expression or the suppression of TAPT1 or Miro1 has the potential to promote local invasion, which increases the risk of metastasis (Figure 5B).

## 3. Discussion

### 3.1. Use of Seed Sequences to Define Novel Regulatory Mechanisms

In this study, we used a high-content screen to identify perturbations that enhance migration through the regulation of gene expression. The miRNA mimics employed directly suppress gene expression by targeting seed sequences present in the 3′ UTRs of genes. In addition, miRNAs can promote indirect secondary shifts in gene expression, fundamentally altering intrinsic cell signaling networks. This ability of miRNA mimics to alter the expression of hundreds of genes creates a pooled approach for perturbing the expression of multiple genes to define regulators of cell phenotypes. Reducing the number of individual conditions to evaluate provides an advantage in high-content screens, such as wound closure assays, that require more time and resource intensive analysis than cell growth assays. Indeed, our results reveal the potential of using miRNA mimics in high-content screens in conjunction with gene expression analysis to prioritize genes for investigation as new regulators of phenotypes such as cell migration. TAPT1, for example, would not be a logical candidate based on standard prioritization approaches, such as testing known enzymes and components of the cytoskeleton [20].

### 3.2. miR614

Our results show that the expression of an miR614 mimic enhances migration and suggests that increased miR614 expression has the potential to promote elements of metastasis in patients. Indeed, miR614 is more highly expressed in African American TNBC (AA-TNBC) patients compared to non-Hispanic White TNBC (NHW-TNBC) patients [21]. However, this prior expression analysis did not identify a function for miR614. The biology of TNBC in AA-TNBC is different from that of NHW-TNBC patients, with AA-TNBC tumors showing increased proliferation and greater risk of developing metastasis [21,22,23]. Our results support further investigation into how the elevated miR614 expression in AA-TNBC tumors contributes to cell migration and invasion. Importantly, defining a function for elevated miR614 expression in AA-TNBC tumors would reveal that unique tumor intrinsic properties in AA-TNBC patients, as opposed to societal factors, contribute to the divergent outcomes in AA-TNBC patients [23]. Moreover, AA-TNBC patients expressing high levels of miR614 may benefit from treatment with miR614 inhibitors. Therapeutics targeting miRNAs are an emerging area of study [24]. The functions of miR614 may extend beyond TNBC. Indeed, miR614 is also a marker of pancreatitis, and shows increased expression in pancreatic tumors compared to matched normal samples [25,26]. miR614 expression is also increased in ovarian cancer patients and a subset of ovarian cancer cell lines, in which miR614 promotes proliferation by suppressing the expression of the phosphatase PPP2R2A [27]. Thus, determining the extent to which miR614 contributes to tumor progression beyond breast cancer warrants further evaluation.

Despite the correlative evidence suggesting that increased miR614 had the potential to promote tumor progression, little was understood regarding miR614 functions and mechanisms of action beyond these limited studies. Our results now reveal that miR614 can promote single cell and collective migration in models of TNBC. This induction of motility correlates with an increase in the expression of the EMT-TF Slug. We have previously found that Slug is required for MCFDCIS wound closure and that exogenous Slug accelerates the transition from DCIS to invasive growth in MCFDCIS xenografts [13]. Thus, our results suggest that miR614, at least in part, increases wound closure through increasing Slug expression. Notably, other canonical EMT-TFs, mesenchymal and epithelial markers were unchanged in the MCFDCIS cells, indicating that miR614 could increase the migration of cells in a hybrid EMT state, as well as a more fully mesenchymal state characteristic in which E-cadherin expression is lost in control cells. We have begun to uncover potential direct and indirect targets of miR614 by showing that the miR614 mimic suppressed a cohort of common genes in MCFDCIS and MDAMB231. Based on seed sequence prediction metrics, miR614 directly targets a subset of the suppressed genes, with the remaining alterations in gene expression being indirect responses to primary target suppression. To the best of our knowledge, this dataset provides the first comprehensive unbiased testing of how miR614 can influence gene expression. Indeed, known targets of miR614 are limited, with the phosphatase regulatory subunit PPP2R2A being the only other functional target evaluated [27]. Our analysis pipeline did not reveal PPP2R2A as an miR614 mimic target in either the MCFDCIS cells or MDAMB231 cells, which could reflect a threshold requirement of miR614 for specific target gene suppression or other contextual factors found within distinct cell lineages. Functional testing indicated that the suppression of Miro1 and TAPT1 by miR614 has the potential to promote migration. Our functional testing showed that the depletion of Miro1 or TAPT1 alone was sufficient to enhance Slug expression and migration. This observation suggests that targeting the expression of either of these genes by miR614 may be sufficient to impact cell migration. The suppression of additional miR614 targets may also promote Slug expression and migration, either alone or in concert with Miro1 and TAPT1 suppression. Indeed, siRNA pools targeting additional genes suppressed by miR614, including NKIRAS2 and RRBP1, also enhanced migration. While they were not prioritized for analysis in this study, NKIRAS2 and RRBP1 are potential inhibitors of migration worthy of future investigation. Together, our results provide functional insight into how the elevated expression of miR614 observed in tumors can contribute to disease progression.

### 3.3. Miro1

Our results indicate that reduced Miro1 expression promotes wound closure. The Miro1 protein is localized to the outer membrane of mitochondria [28]. Miro1, and the related Miro2 protein, promote anterograde trafficking of mitochondria by interacting with proteins that bind to microtubule motors and the actin cytoskeleton [29]. Miro1 is also required for symmetrical inheritance of mitochondria during cell division [30]. These actions have been most extensively investigated in motor neurons, which require the long range transport of mitochondria along axons [31]. However, the coordination of mitochondrial trafficking by Miro1 is evolutionarily conserved and functions in non-neuronal cell types [32]. Thus, Miro1 potentially influences the properties of a wide range of tumors. Miro1 is also required for symmetrical inheritance of mitochondria during cell division. Interestingly, depleting Miro1 increased Slug expression. This suggests that miR614 increases Slug expression, at least in part, through suppressing Miro1. This suppression is likely mediated by miR614 directly interacting with a seed sequence of the 3′ UTR of Miro. Mitochondrial disfunction activates retrograde signaling pathways from the mitochondria to the nucleus that induce changes in gene expression [33]. Indeed, calcium mediated retrograde signaling can induce an EMT and Slug expression [34]. Miro1 depletion alters the subcellular distribution of mitochondria, perturbs mitophagy and Miro can regulate mitochondrial morphology [28]. Whether these alterations in mitochondrial function resulting from Miro1 loss also activate retrograde signaling to promote Slug expression is an interesting line of future investigation. Defining the specific mechanism of Slug regulation by Miro1 will assist in determining the relative contribution of Slug towards inducing the migration of cells with reduced Miro1 expression. Understanding how Miro1 loss promotes Slug expression and migration may also reveal new treatment options.

The increased migration of cells resulting from Miro1 suppression may also be dependent on cell context. Miro1 loss reduces the migration rate in fibroblasts and leukocytes [35,36]. This reduced migration may be a consequence of Miro1 depletion, leading to subtle perturbations in the subcellular ratio of ATP to ADP that alter cell adhesion dynamics [35,36]. One possibility is that the second Miro family member Miro2 [28] is sufficient to maintain ATP/ADP ratios in our model whereas other cell types are more reliant on Miro1. The relative expression of Miro1 and Miro2 does vary between cell types [31]. The changes in gene expression induced by Miro1 loss, including Slug expression, may also have a compensatory effect.

The removal of Miro1 from the outer mitochondrial membrane is necessary for the clearance of damaged mitochondria through mitophagy [31]. The phosphorylation of Miro1 by PINK1 promotes the Parkin mediated ubiquitination and subsequent degradation of Miro1 by the proteasome [37]. This degradation of Miro1 prevents the trafficking of damaged mitochondria and is the first step that results in mitophagy [38]. Indeed, mutations in PINK1 and Parkin delay the removal of Miro1 from damaged mitochondria and cause Parkinson’s disease [39]. Whether the suppression of Miro1 expression by miR614 has the converse effect of accelerating mitophagy, or whether miR614 could be used to normalize mitophagy in Parkinson’s disease patients, is an interesting line of future investigation.

### 3.4. TAPT1

miR614 suppressed TAPT1 expression and the depletion of TAPT1 was sufficient to enhance the rate of wound closure. TAPT1 is a universally expressed and evolutionarily conserved protein with multiple predicted transmembrane domains [40]. Mutations in TAPT1 are associated with developmental defects in zebrafish, mice and humans, including skeletal defects and perturbations in the development of the brain, kidney and lungs [40,41]. In humans, these familial TAPT1 mutations shift the localization of TAPT1 from the centrosome to cytoplasm, which may contribute to the disruption in cilia formation, Golgi structure and protein secretion observed in TAPT1 mutant cells [41]. These combined findings reveal an essential role for TAPT1 during development. However, TAPT1 interacting proteins have yet to be identified and little is understood with respect to TAPT1’s function beyond this initial characterization of defects that are associated with familial or experimentally induced TAPT1 mutations. Cilia contribute to the regulation of intracellular signaling pathways [42], raising the possibility that the increased rate of migration and Slug expression in TAPT1 depleted cells is a consequence of disruptions of cilia. It is also possible that TAPT1 has yet to be described functions that restrict migration and the level of Slug in cells. Determining how TAPT1 restricts Slug expression in the future will assist in understanding the influence of Slug regulated programs versus other signaling pathways in promoting migration in response to increased miR614 expression or TAPT1 suppression. This additional insight into how the loss of TAPT1 promotes migration may also reveal options for therapeutic intervention.

The lack of an miR614 seed sequence in TAPT1 suggests that miR614 indirectly regulates TAPT1 expression. A single siRNA targeting LAMC1 reduces TAPT1 expression [43]. However, this observation has not been confirmed by additional LAMC1 depletion approaches and LAMC1 was not suppressed by miR614 in MCFDCIS or MDAMB231 cells. TAPT1 expression is suppressed by polycomb repressive complex 2 (PRC2) in leukemia cells, although the functional significance of TAPT1 suppression was not determined [44]. Whether miR614 activates PRC2 to suppress TAPT1 expression or acts through a separate mechanism requires further investigation. Notably, further inquiry into TAPT1 regulation and function is supported by the correlation between reduced TAPT1 expression and worse patient outcomes in breast cancer and NSCLC patients. Given the ubiquitous expression of TAPT1, it may act as suppressor of migration and serve as a biomarker for patient outcome in additional tumor types.

In summary, using a data driven discovery approach, we have uncovered new regulator mechanisms underpinning the control of cell migration and a potential biomarker for stratifying patients into outcome groups.

## 4. Materials and Methods

### 4.1. Cell Culture

MCFDCIS (Asterand, Detroit, MI, USA) and MDAMB231 (ATCC, Manassas, VA, USA) cells were cultured as described [45] at 5% CO_2_ and 37 °C. MCFDCIS growth medium was DMEM-F12 supplemented with 5% horse serum, 20 ng/mL EGF, 0.5 µg/mL hydrocortisone, 100 ng/mL cholera toxin, 10 µg/ml insulin and 1X pen/strep. MDAMB231 cells growth medium was RPMI supplemented with 10% fetal bovine serum (FBS) and 1× pen/strep. The generation of the MCFDCIS and MDAMB231 cell lines stably expressing H2B:GFP was previously described [45].

### 4.2. Immunofluorescence

Fixation was performed by incubating cells for 20 min in 2% formaldehyde. Fixed cells were then permeabilized for 10 min using 0.05% Triton X-100. After blocking with 10% goat serum in IF buffer (0.1% BSA, 0.2% Triton X-100, 0.05% Tween-20 in PBS) for 1 h, cells were immunostained with E-cadherin antibody overnight and counter stained with Phalloidin-546 (Life Technologies, Waltham, MA, USA, A22283) and 1:2000 Hoechst (Life Technologies, H1399) for 1 h. See Appendix A for details. Plates were stored at 4 °C until imaged.

### 4.3. Transfection of siRNAs and miRNAs

Transfections were performed as described previously [19]. Either siRNAs (5–50 nM) or miRNA mimics (10–50 nM) were transfected into target cells using RNAiMax (Invitrogen, Waltham, MA, USA, 13,778,500). For experiments testing siRNAs, a pool of siRNAs that do not target human genes was used as a negative control. For experiments testing miRNA mimics, a mimic for miR545, which produced no phenotype compared to mock transfected cells (no siRNA) or control siRNA transfected cells [13], was used as a negative control. Details of the siRNAs and miRNA mimics are located in Appendix A.

### 4.4. Wounding Assays

A 96-pin wounding tool (AFIX96FP6, V&P Scientific, San Diego, CA, USA) was used to generate reproducible experimental wounds across replicates in all experiments, as we have done previously [13]. The wounding tool contains 1.68 mm diameter pins (FP6-WP). A template holder was used to generate wounds with a length of 4.5 mm (VP 381NW 4.5, V&P Scientific). Immediately after wounding, wells were washed twice with media to remove cellular fragments after wounding. Cells were allowed to migrate for 24 h after wounding and were then fixed in 2% formaldehyde. The fixed cells were stained with Hoechst (Life Technologies, H1399) and phalloidin-546 (Life Technologies, A22283). Imaging was performed using a BD Pathway 855 microscope with a 10× objective (Olympus, UPlanSApo 10×/0.40, ∞/0.17/FN26.5). Images were acquired as 4 × 5 or 6 × 4 montages. Quantification of wound closure was performed using Pipeline Pilot software, which defined the amount of empty space in each well not occupied by cells using a threshold of pixel signal intensity. The Hoechst signal detected using a Pherastar plate reader was used to determine the relative number of cells per well.

### 4.5. Time-Lapse Imaging

Imaging of cells expressing H2B:GFP in 96-well glass bottom plates was performed at 30 min intervals for 7 h total using a Perkin Elmer Ultraview ERS spinning disk confocal microscope with a CCD camera (Orca AG; Hamamatsu, Hamamatsu City, Shizuoka, Japan). The system was housed in an incubation chamber set to 37 °C and cells were also provided with humidified CO_2_ (Solent, Portsmouth, UK). At least 5 x,y points per condition were imaged. A 10× (Zeiss, Oberkochen Germany) objective using Volocity software (Perkin Elmer, Waltham, MA, USA) for 7 h at 30 min intervals in each experiment with Imaris software was used to quantify cell speed and displacement, as described [45].

### 4.6. Transwell Migration Assays

Twenty-four hours after transfection, 20,000 MDAMB231-H2B:GFP were plated onto transwell filters (BD Falcon Cell Culture Inserts, Tewksbury, MA, USA, 353,097) in 1% FBS/RPMI media. Transwell migration towards a reservoir of 10% FBS/RPMI for 6 h was then measured. A cotton swab was used to remove non-migrating cells from the upper side of the transwell filter. An inverted fluorescence microscope was used to quantify the number of cells that migrated through the transwell filter pores, as described previously [46].

### 4.7. Quantitative Real-Time PCR

RNAeasy purification columns (Qiagen) were used to isolate total mRNA and iScript cDNA Synthesis Kits (Bio-Rad, Hercules, CA, USA) were used to generate cDNA. The primers described in Appendix A were used to perform TaqMan Gene Expression Assays from 20 ng of cDNA using an Applied Biosystems 7500 Real-Time PCR System. The ∆∆CT method was applied to quantify relative gene expression compared to GAPDH from triplicate samples as described [47].

### 4.8. Gene Expression Profiling

Human HT-12 v4 Expression BeadChips (Illumina, San Diego, CA, USA) were used to measure genome-wide mRNA expression. The data were then processed as described [48] and quantile–quantile normalized. Differentially expressed genes were determined by fold-change and the reproducibility of replicate samples. Heatmaps showing the relative expression of genes were generated with pHeatmap using R version 3.5.1. The mRNA expression data are available at the GEO (GSE159,639).

### 4.9. Patient Survival Analysis

The KM-plotter meta-analysis database [49] available at KMplot.com (Budapest, Hungary) was used to determine the survival of breast cancer and NSCLC patients. Patients were stratified into “high” and “low” groups based on the upper tertile of gene expression. Survival differences were compared by log-rank test. The TAPT1 probe 238,798 and Miro1 probe 218,323 were used.

### 4.10. Statistical Methods

Two-tailed Student’s *t*-tests (Graphpad Prism, San Diego, CA, USA) were used to analyze all experimental data. The patient survival differences were analyzed using log-rank tests. *p*-values < 0.05 were considered significant.

## Figures and Tables

**Figure 1 ijms-22-00112-f001:**
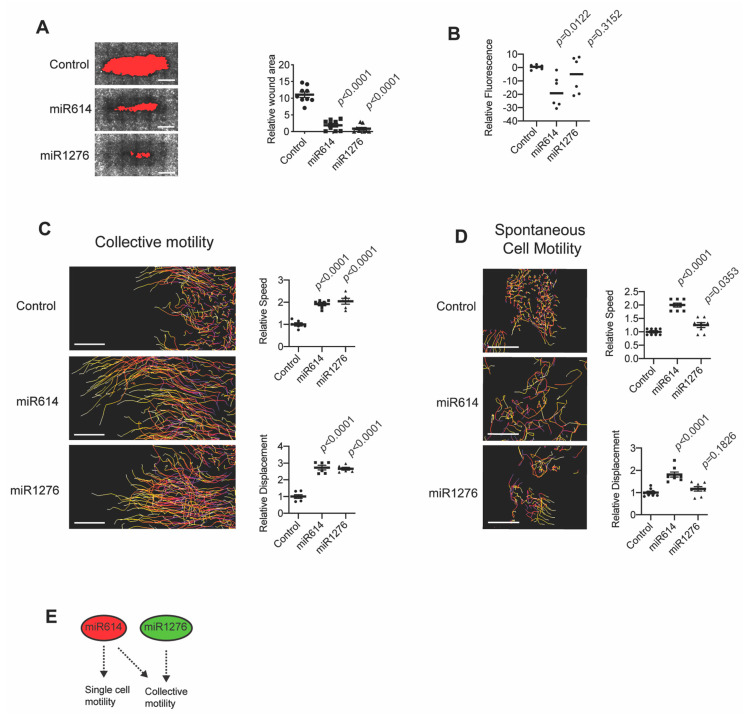
miR614 enhances migration. (**A**) Representative images of MCFDCIS cells stained with phalloidin showing that transfection of miR614 or miR1276 mimics enhances the rate of wound closure over 24 h. The wound area, indicated in red, is determined by fluorescent signal threshold analysis that defines cell-free space. Graph shows the quantification of wound area. Less wound area indicates faster migration. *n* = 9 wells imaged in 3 independent experiments. Scale bars, 1 mm. (**B**) Transfection of miR614 or miR1276 does not increase proliferation. Graph shows the fluorescence intensity of Hoechst stained cells after completion of a 24 h wound closure assay. *n* = 6 wells from 2 independent experiments. (**C**) MCFDCIS cells transfected as indicated were wounded and imaged for 7 h. Tracking of cell movement is shown. Color changes indicate increasing time. Scale bars, 100 µm. Graphs show quantification of cell speed and displacement (mean ± SD, *n* = 6 x,y positions over 2 independent experiments). (**D**) Subconfluent MCFDCIS cells transfected as indicated were imaged for 7 h. Tracking of cell movement is shown. Color changes indicate increasing time. Scale bars, 100 µm. Graphs shows quantification of cell speed displacement (mean ± SD, *n* = 6 x,y positions over 2 independent experiments). (**E**) Model showing that miR614 promotes single cell and collective migration while miR1276 specifically promotes collective migration. *p*-values determined by unpaired Student’s *t*-test.

**Figure 2 ijms-22-00112-f002:**
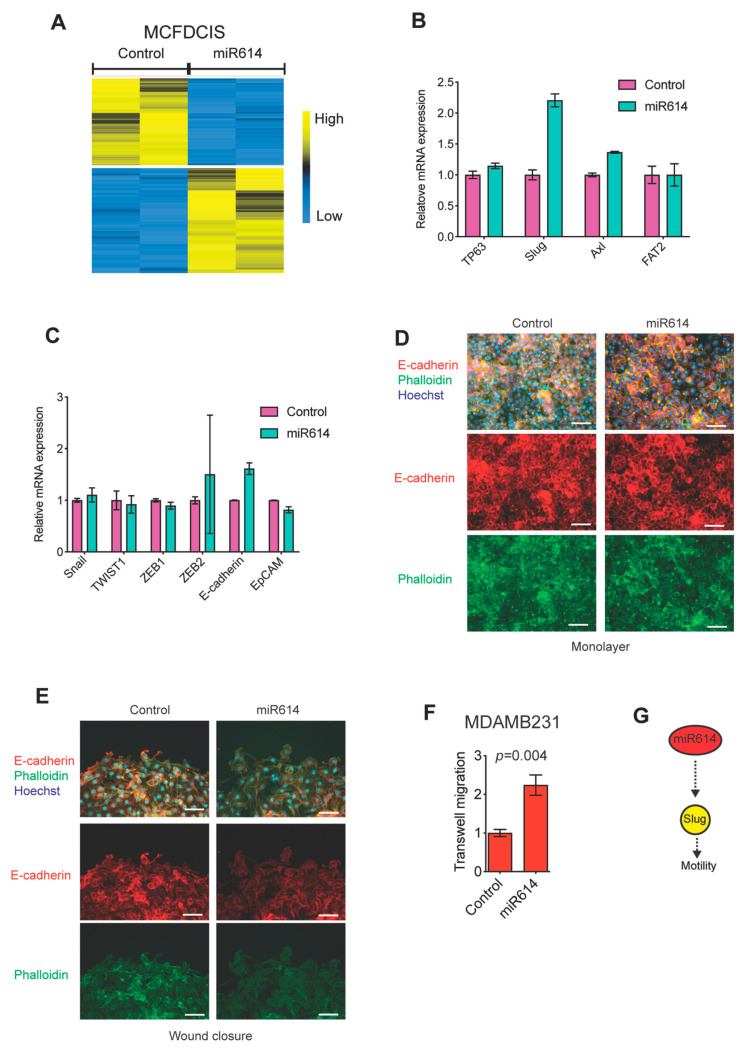
miR614 induces Slug expression. (**A**) Heatmap showing how miR614 alters gene expression in MCFDCIS cells. Biological replicates are shown. Yellow indicates increased expression and blue indicates reduced expression. (**B**) Graph shows how miR614 influences the expression of genes that have previously been shown to be required for MCFDCIS cell migration. (**C**) Graph shows how miR614 influences the expression of epithelial-to-mesenchymal transition (EMT) inducing transcription factors and the epithelial genes E-cadherin (CDH1) and EpCAM. (**D**) Representative images of confluent MCFDCIS cells 72 h after transfection. Cells were immunostained with anti-E-cadherin antibody (red) and counterstained with Hoechst (blue, nuclei) and phalloidin (green, F-actin). Scale bars, 50 µm. (**E**) Representative images of migrating MCFDCIS cells after wounding. Cells were transfected as indicated, immunostained with anti-E-cadherin antibody (red) and counterstained with Hoechst (blue, nuclei) and phalloidin (green, F-actin). Scale bars, 25 µm. (**F**) miR614 increases the transwell migration of MDAMB231 cells. MDAMB231 cells were transfected with control or miR614 mimic for 48 h and then re-plated onto transwell inserts in serum free media. Transfected cells were then allowed to migrate towards serum containing media for 24 h. Graph shows the number of cells that migrated through the insert divided by the total number of cells on the top and underside of the insert and normalized to the control. Mean ± SD of 3 independent experiments. *p*-values determined by unpaired Student’s *t*-test. (**G**) Model showing that miR614 increases the expression of Slug and cell motility.

**Figure 3 ijms-22-00112-f003:**
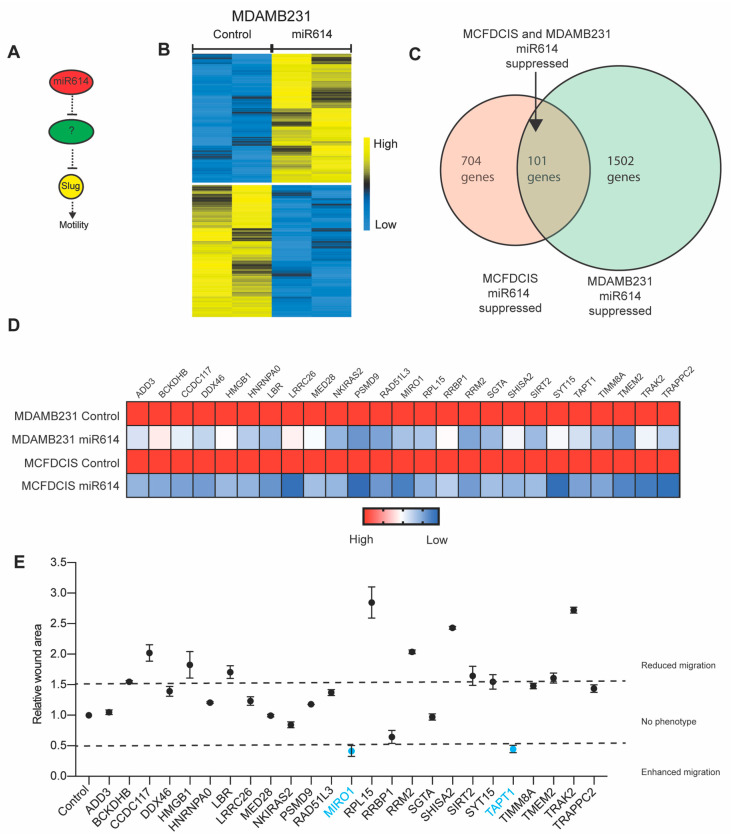
Defining genes suppressed by miR614 that restrict cell migration. (**A**) Model showing that miR614 indirectly regulates the expression of Slug and cell motility through the suppression of gene expression. (**B**) Heatmap showing how miR614 alters gene expression in MDAMB231 cells. Biological replicates are shown. Yellow indicates increased expression and blue indicates reduced expression. (**C**) Venn diagram showing how miR614 suppresses a common set of genes in MCFDCIS and MDAMB231 cells. (**D**) Heatmap showing the suppression of 27 genes by miR614 in MCFDCIS and MDAMB231 cells. All genes had a minimum probe value of 50 when analyzed by microarray. (**E**) Graph showing the results of a screen in which the wound area of MCFDCIS cells transfected with siRNAs targeting the 27 miR614 suppressed genes shown in (**D**) was determined. The mean (solid circles) ± SD of 3 wells for each condition tested is shown. Dashed lines indicate the threshold for a 50% change in wound closure. The results of siRNAs for targeting Miro1 and TAPT1 are indicated in blue and were prioritized for further investigation.

**Figure 4 ijms-22-00112-f004:**
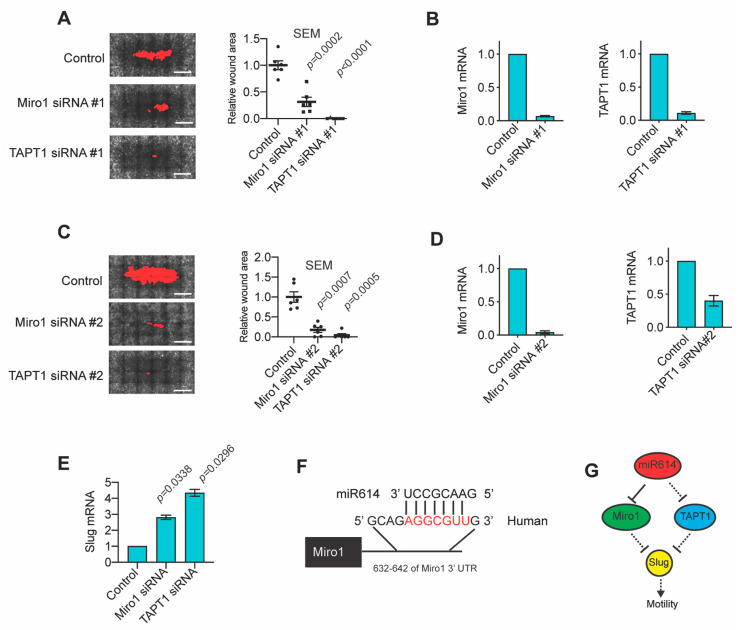
The miR614 suppressed genes Miro1 and TAPT1 are inhibitors of migration and Slug expression. (**A**) Representative images showing that transfection of MCFDCIS cells with siRNAs targeting Miro1 or TAPT1 enhances wound closure. Wound area is shown in red. Graph shows the mean ± SEM of 6 wells analyzed. Scale bars, 1 mm. (**B**) Graphs showing that the expression of Miro1 and TAPT1 are suppressed by their respective siRNA pools and miR614. (**C**) Representative images showing that transfection of MCFDCIS cells with a second distinct set of siRNAs targeting Miro1 or TAPT1 enhances wound closure. Wound area is shown in red. Graph shows the mean ± SEM of 6 wells analyzed. Scale bars, 1 mm. (**D**) Graphs showing that the second distinct sets of Miro1 and TAPT1 siRNAs suppress target gene expression. (**E**) Graph showing that siRNAs targeting Miro1 or TAPT1 increase Slug expression. (**F**) Model showing the predicted seed sequence in the 3′ UTR of Miro1 that is targeted by miR614. (**G**) Model showing that miR614 directly suppresses Miro1 and indirectly suppresses TAPT1. The suppression of these genes by miR614 increases Slug expression and cell motility.

**Figure 5 ijms-22-00112-f005:**
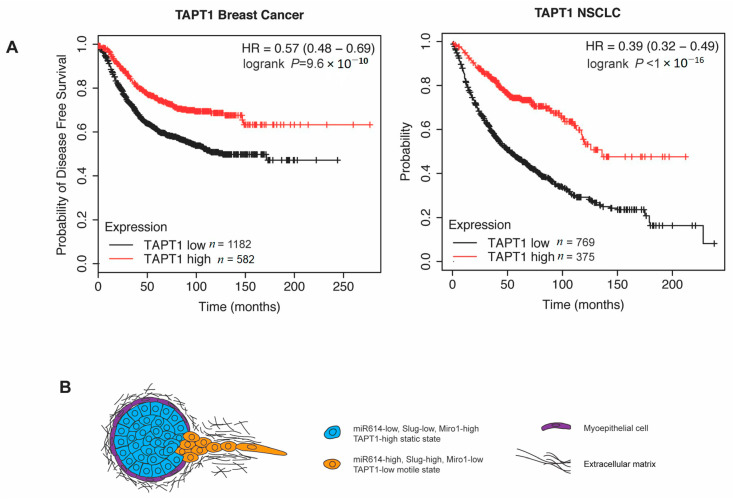
Reduced TAPT1 expression correlates with worse outcome in breast cancer and non-small cell lung cancer (NSCLC) patients. (**A**) Kaplan–Meier curves showing the disease-free survival of breast cancer patients and NSCLC patients classified as “TAPT1 high” and “TAPT1 low” based on TAPT1 mRNA expression. Survival differences were compared by log-rank test. (**B**) Model proposing how miR614 promotes a more motile EMT state by inhibiting Miro1 and TAPT1 expression.

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
