# Peer review of "miR614 Expression Enhances Breast Cancer Cell Motility"

_ijms, 2020, doi:10.3390/ijms22010112_

Round 1
Reviewer 1 Report
In this work, the authors characterize the role of miR614 in the regulation of EMT state and migratory properties of cancer cells. The manuscript shows a nice job and the results have a wide interest in cancer research. Prior to acceptance, a major concern should be considered by the authors.
Major point
The authors claim that “miR614 expression enhances cell motility through suppression of Miro1 and TAPT1” but this statement is based in experimental correlations. For this statement to be consistent, additional genetic experiments are required. For instance, in wound healing experiments, the miR614 and Slug siRNA (Dang et al, Cancer Research, 2015) combination should inhibit wound closure. Also, Miro1 or TAPT1 siRNA and Slug siRNA combination should difficult wound closure. Probably, those are the easiest experiments to circumvent the use of Miro1 and TAPT1 overexpression. Although the results are predictable and seems obvious based on the other results obtained, this information is relevant to maintain the claim proposed in the title of the paper.
Minor points
1- in line 276, “of” instead of “if”
2- For the data in figure 5a and S2, it seems that the reference and the link to the used database are missing.
Author Response
Thank you for the suggestion. We believe that our results showing that the depletion of the
miR614 targets Miro1 and TAPT1 induces Slug expression and increases migration provides a
plausible mechanism by which miR614 can induce migration. We agree that the depletion of
Slug will perturb the increased migration induced by miR614, or the depletion of Miro1 or TAPT
based on our previous reports. However, the advance provided by this experiment is limited by
the fact that it will not distinguish between a general requirement of Slug induced by any factor
versus a specific response triggered by a pathway that regulates Slug expression.
The ability of miR614 to potentially enhance migration through multiple target genes that
converge to regulate the same signaling pathway and cell phenotype is intriguing. However, it
also complicates interpreting experiments in which Miro1 or TAPT1 are exogenously expressed
in cells transfected with a miR614. Based on our results showing that depletion of Miro1 and
TAPT1 alone is sufficient to induce Slug and migration, it is likely that Miro1 and TAPT1 will both
need to be expressed simultaneously, at minimum, to prevent miR614 from enhancing
migration. In addition, it is possible that miR614 regulates additional signaling pathways the
promote migration in parallel to the regulation of Slug, as is suggested by our results showing
that siRNAs targeting at least 2 additional miR614 targets, RRBP1 and NKIRAS2 reproducibly
enhanced migration.
Given the extensive additional analysis that would be be required, we believe that the further
delineation into the specific contribution of Miro1 and TAPT1 towards cell properties regulated
by miR614 is beyond the scope of this investigation. We have made revisions to the title and
clarifications in the text that we believe are consistent with the data provided in this manuscript
showing that miR614 and the miR614 target genes Miro1 and TAPT1 regulate Slug expression
and breast cancer cell migration.
The title has been revised to “miR614 expression enhances breast cancer cell motility”.
Revisions to text that clarify our results and describe the future challenges of defining miR614,
Miro1 and TAPT1 function are found in the following locations in the manuscript:
Lines 87-88, 267-269, 291, 308-309, 368-373, 409-411, 446, 464-466.
Minor points
1- in line 276, “of” instead of “if”
Thank you for making us aware of this error. The typo has been corrected. The correct use of
“of” is part of line 308 of the revised manuscript.
2- For the data in figure 5a and S2, it seems that the reference and the link to the used
database are missing.
The website (KMPlot.com), link and reference are on line 551 of the revised manuscript.
Reviewer 2 Report
Dang, Pearson et al. address an interesting question analyzing the impact of miR614 expression on cell motility and the respective modalities. This study focuses on factors affecting epithelial-to-mesenchymal transition-driven cell migration, building up on their previous would closure model examining miRNAs that provoked the greatest enhancement of wound closure.
The authors mention that miR614 expression is also increased in ovarian cancer patients and a subset of ovarian cancer cell lines, but similar or contradictory findings are stated for other malignant entities or Parkinson’s disease, which are highly dependent on EMT.
There are a few issues that should be clarified/addressed by the authors:
1) Overall low number of references, missing some of the recent most significant publications regarding Miro1 and TAPT1, such as:
Grossmann D, Berenguer-Escuder C, Chemla A, Arena G, Krüger R. The Emerging Role of RHOT1/Miro1 in the Pathogenesis of Parkinson's Disease. Front Neurol. 2020 Sep 15;11:587. doi: 10.3389/fneur.2020.00587. PMID: 33041957; PMCID: PMC7523470.
Bharat V, Wang X. Precision Neurology for Parkinson's Disease: Coupling Miro1-Based Diagnosis with Drug Discovery. Mov Disord. 2020 Jul 25. doi: 10.1002/mds.28194. Epub ahead of print. PMID: 32710675.
Kunitomi H, Kobayashi Y, Wu RC, Takeda T, Tominaga E, Banno K, Aoki D. LAMC1 is a prognostic factor and a potential therapeutic target in endometrial cancer. J Gynecol Oncol. 2020 Mar;31(2):e11. doi: 10.3802/jgo.2020.31.e11. Epub 2019 Aug 21. PMID: 31912669; PMCID: PMC7044014.
Please include the above-mentioned citations in the discussion and critically address them.
2) Further elaboration of the role of Miro1 in association with Parkinson’s disease is required in the Discussion section.
3) A more versatile analysis of translational and clinical implications in various tumor entities is required in the discussion section.
4) A professional proof-reading of the revised manuscript is strongly recommended prior to resubmission.
Author Response
Thank you for making us aware of these references. Nine additional references have been
added, including the 3 suggested references. Specifically, Grossmann et. al. is ref. 38 on line
426; Bharat et. al. is ref. 39 on line 427; Kunitomi et. al. is ref. 43 on line 469.
2) Further elaboration of the role of Miro1 in association with Parkinson’s disease is required in
the Discussion section.
Thank you for the suggestion. A paragraph describing the function of Miro1 in Parkinson’s
disease is included from lines 422-430. This paragraph includes the suggested Grossmann et.
al. and Bharat et. al. references.
3) A more versatile analysis of translational and clinical implications in various tumor entities is
required in the discussion section.
We appreciate the interest in the translation implications of our study.
The translation implications for miR614 are described on lines 325-345.
The translation implications for Miro1 are described on lines 398-399, 408-411 and 427-430.
The translation implications for TAPT1 are described on lines 445-468 and 473-478.
4) A professional proof-reading of the revised manuscript is strongly recommended prior to
resubmission.
Thank you for the suggestion. The authors have each proof-read the revised manuscript. We
believe that the typos have been corrected.
Round 2
Reviewer 1 Report
The authors have clarified the results better and the change in title is more in line with the conclusions. Now, the article is acceptable for publication in IJMS.